# A Technical and Normative Investigation of Social Bias Amplification

## Abstract

The conversation around the fairness of machine learning models is growing and evolving. In this work, we focus on the issue of *bias amplification*: the tendency of models trained from data containing social biases to further amplify these biases. This problem is brought about by the algorithm, on top of the level of bias already present in the data. We make two main contributions regarding its measurement. First, building off of Zhao et al. (2017), we introduce and analyze a new, decoupled metric for measuring bias amplification, $BiasAmp_{\rightarrow}$, which possesses a number of attractive properties, including the ability to pinpoint the cause of bias amplification. Second, we thoroughly analyze and discuss the normative implications of this metric. We provide suggestions about its measurement by cautioning against predicting sensitive attributes, encouraging the use of confidence intervals due to fluctuations in the fairness of models across runs, and discussing what bias amplification means in the context of domains where labels either don't exist at test time or correspond to uncertain future events. Throughout this paper, we work to provide a deeply interrogative look at the technical measurement of bias amplification, guided by our normative ideas of what we want it to encompass.

## 1 Introduction

The machine learning community is becoming increasingly cognizant of problems surrounding fairness and bias, and correspondingly a plethora of new algorithms and metrics are being proposed (see e.g., Mehrabi et al. (2019) for a review). The gatekeepers checking the systems to be deployed often take the form of fairness evaluation metrics, and it is vital that these be deeply investigated both technically and normatively. In this paper, we endeavor to do this for bias amplification. Bias amplification happens when a model exacerbates biases from the training data at test time. It is the result of the algorithm (Foulds et al., 2018), and unlike other forms of bias, cannot be solely attributed to the dataset.

To this end, we propose a new way of measuring bias amplification, $BiasAmp_{\rightarrow}$[1], that builds off a prior metric from Men Also Like Shopping (Zhao et al., 2017), that we will call $BiasAmp_{MALS}$. Our metric's technical composition aligns with the real-world qualities we want it to encompass, addressing a number of the previous metric's shortcomings by being able to: 1) generalize beyond binary attributes, 2) take into account the base rates that people of each attribute appear, and 3) disentangle the directions of amplification. Concretely, consider a visual dataset (Fig. 1) where each image has a label for the task, $T$, which is painting or not painting, and further is associated with a protected attribute, $A$, which is woman or man. If the gender of the person biases the prediction of the task, we consider this $A \rightarrow T$ bias amplification; if the reverse happens, then $T \rightarrow A$.

In our normative discussion, we discuss a few topics. We consider whether predicting protected attributes is necessary in the first place; by not doing so, we can trivially remove $T \rightarrow A$ amplification. We also encourage the use of confidence intervals when using our metric because $BiasAmp_{\rightarrow}$, along with other fairness metrics, suffers from the Rashomon Effect (Breiman, 2001), or multiplicity of good models. In deep neural networks, random seeds have relatively little impact on accuracy; however, that is not the case for fairness, which is more brittle to randomness.

---

[1]The arrow in $BiasAmp_{\rightarrow}$ is meant to signify the direction that bias amplification is flowing, and not intended to be a claim about causality.

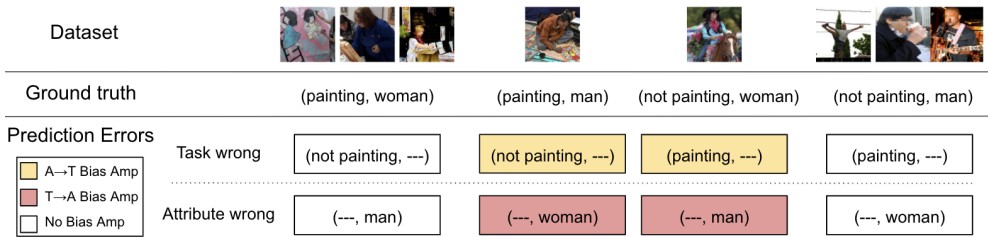

Figure 1: Consider an image recognition dataset where the goal is to classify the task, $T$, as painting or not painting, and the attribute, $A$, as woman or man. Note that in this dataset women are correlated with painting, and men with not painting. In this work we are particularly concerned with errors that contribute to the amplification of bias (red and yellow in the figure), i.e., those that amplify the training correlation. We further disentangle these errors into those that amplify the attribute to task correlation (i.e., incorrectly predict the task based on the attribute of the person; shown in yellow) versus those that in contrast amplify the task to attribute correlation (shown in red).

Notably, a trait of bias amplification is that it is not at odds with accuracy, unlike many other fairness metrics, because the goal of not amplifying biases and matching task-attribute correlations is aligned with that of accurate predictions. For example, imagine a dataset where the positive outcome is associated at a higher rate with group A than with group B. A classifier that achieves 100% accuracy at predicting the positive outcome is not amplifying bias; however, according to metrics like demographic parity, this perfect classifier is still perpetuating bias because it is predicting the positive label at different rates for both groups. While matching training correlations is desired in object detection where systems should perfectly predict the labels, we will explore the nuances of what this means in situations where the validity of the labels, and thus task-attribute correlations themselves, are up for debate. For example, in the risk prediction task which assesses someone's likelihood of recidivism, the label represents whether someone with a set of input features ended up recidivating, but is not a steadfast indicator of what another person with the same input features will do. Here, we would not want to replicate the task-attribute correlations at test time, and it is important to keep this in mind when deciding what fairness metrics to apply. The notion of amplification also allows us to encapsulate the idea that systemic harms and biases can be more harmful than errors made without such a history (Bearman et al., 2009); for example, in images overclassifying women as cooking carries more of a negative connotation than overclassifying men as cooking.[2] Distinguishing between which errors are more harmful than others is a pattern that can often be lifted from the training data.

To ground our work, we first distinguish what bias amplification captures that standard fairness metrics cannot, and then distinguish BiasAmp$_\rightarrow$ from BiasAmp$_{\text{MALS}}$. Our key contributions are: 1) proposing a new way to measure bias amplification, addressing multiple shortcomings of prior work and allowing us to better diagnose where a model goes wrong, and 2) providing a technical analysis and normative discussion around the use of this measure in diverse settings, encouraging thoughtfulness with each application.

## 2  RELATED WORK

**Fairness Measurements.** Fairness is nebulous and context-dependent, and approaches to quantifying it (Verma & Rubin, 2018; Buolamwini & Gebru, 2018) include equalized odds (Hardt et al., 2016), equal opportunity (Hardt et al., 2016), demographic parity (Dwork et al., 2012; Kusner et al., 2017), fairness through awareness (Dwork et al., 2012; Kusner et al., 2017), fairness through unawareness (Grgic-Hlaca et al., 2016; Kusner et al., 2017), and treatment equality (Berk et al., 2017). We examine bias amplification, which is a type of group fairness where correlations are amplified.

**Bias Amplification.** Bias amplification has been measured by looking at binary classifications (Leino et al., 2019), GANs (Jain et al., 2020; Choi et al., 2020), and correlations (Zhao et al., 2017). Wang et al. (2019) measures this using dataset leakage and model leakage. The difference between these values is the level of bias amplification, but this is not a fair comparison because the

---

[2]We use the terms man and woman to refer to binarized socially-perceived gender expression, recognizing these labels are not inclusive, and in vision datasets are often assigned by annotators rather than self-disclosed.

attribute classifier gets discrete labels for the former but continuous model outputs for the latter. Jia et al. (2020) looks at output distributions like we do, but with a different formulation.

The Word Embedding Association Test (WEAT) (Caliskan et al., 2017) measures bias amplification in de-contextualized word embeddings, looking at correlations but not causations (Bolukbasi et al., 2016). However, with newer models like BERT and ELMo that have contextualized embeddings, WEAT does not work (May et al., 2019), so new techniques have been proposed incorporating context (Lu et al., 2019; Kuang & Davison, 2016). We use these models to measure the directional aspect of these amplifications, as well as to situate them in the broader world of bias amplification.

**Directionality.** Directionality of amplification has been observed in computer vision (Stock & Cisse, 2018) and language (Qian et al., 2019). It has also been studied with causality (Bhattacharya & Vogt, 2007; Wooldridge, 2016; Pearl, 2010; Middleton et al., 2016; Steiner & Yongnam, 2016). We take a deeper and more empirical approach.

**Predictive Multiplicity.** The Rashomon Effect (Breiman, 2001), or multiplicity of good models, has been studied in various contexts. The variables investigated that differ across good models include explanations (Hancox-Li, 2020), individual treatments (Marx et al., 2020; Pawelczyk et al., 2020), and variable importance (Fisher et al., 2019; Dong & Rudin, 2019). We build on these discoveries and investigate how fairness measurements also differ between equally "good" models.

## 3 EXISTING FAIRNESS METRICS

In this section we present existing fairness metrics and show how bias amplification can distinguish errors resulting from under- and overclassification in a way that others cannot, followed by a discussion of the shortcomings of BiasAmp$_{\text{MALS}}$.

### 3.1 OVERVIEW OF EXISTING FAIRNESS METRICS

We begin with a review of existing fairness metrics in a concrete classification setting. We consider again the example from Fig. 1, where on this dataset women ($a_0$) are correlated with painting ($t_0$), and men ($a_1$) with not painting ($t_1$), such that there are $N$ images each of ($a_0, t_0$) and ($a_1, t_1$) but only $N/2$ images of ($a_0, t_1$) and ($a_1, t_0$). A classifier trained to recognize painting on this data is likely to learn this association and over-predict painting on images of women and under-predict painting on images of men; however, algorithmic interventions may counteract this effect and in fact result in the opposite behavior.

Fig. 2 shows the behavior of fairness metrics under varying amounts of learned amplification of the correlation. The four fairness metrics are: *False Positive Rate (FPR) and True Positive Rate (TPR) difference:* the difference in false positive (true positive) rate of predicting the label $t_0$ on images of $a_1$ versus on images of $a_0$ (Chouldechova, 2016; Hardt et al., 2016), *accuracy difference:* difference between the overall task prediction accuracy on images of $a_1$ versus on images of $a_0$ (Berk et al., 2017), and *mean accuracy across subgroups:* mean task prediction accuracy across the four image subgroups (($a_0, t_0$), ($a_1, t_0$), ($a_0, t_1$), and ($a_1, t_1$)) (Buolamwini & Gebru, 2018). However, these metrics are not designed to account for the training correlations, and are unable to distinguish between cases of increased or decreased learned correlations, as seen in Fig. 2.

Zhao et al. (2017) introduced an alternative to these that explicitly captures the notion of bias amplification. Concretely, they consider $P(A = a | T = t)$ of the training data as the fraction of times a protected attribute $a \in \mathcal{A}$ appears on images corresponding to task $t \in \mathcal{T}$. They then compare this with the test-time predictions made by the model, $P(\hat{A} = a | \hat{T} = t)$, or the number of times attribute $a$ is predicted on images where the task is predicted as $t$, which allows them to measure bias amplification in the absence of any additional annotations on the hold-out set. Note that in this formulation they are assuming that the model is making predictions for both the task and the attribute. The full bias amplification metric (reformulated in our terms), is computed as BiasAmp$_{\text{MALS}}$ =

$$\frac{1}{|\mathcal{T}|} \sum_{t=1}^{|\mathcal{T}|} \sum_{a=1}^{|\mathcal{A}|} \underbrace{\mathbb{1}\left(P(A_a = 1|T_t = 1) > \frac{1}{|\mathcal{A}|}\right)}_{y(t,a)} \underbrace{\left(P(\hat{A}_a = 1|\hat{T}_t = 1) - P(A_a = 1|T_t = 1)\right)}_{\Delta(t,a)} \quad (1)$$

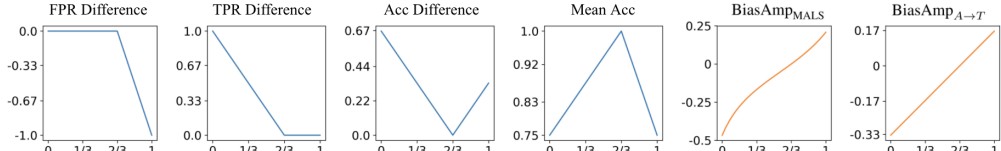

Figure 2: Different fairness metrics vary in how they respond to model errors. In our image dataset (Fig. 1) of predicting someone who is a woman or man to be painting or not, we consider a classifier that always predicts the task correctly for men, but varies for women. The x-axes of the graphs correspond to the fraction of women predicted to be painting, where the ground-truth is $2/3$, and the model does not predict false positives before this point. The first two metrics, FPR and TPR difference, only capture one of under- or overclassification. The next two metrics are symmetric around $2/3$, being unable to differentiate under- or overclassification. Both bias amplification metrics are able to distinguish between under- and overclassification.

Fig. 2 empirically demonstrates that this metric is able to capture the level of increasing bias amplification. (For consistency in comparison with prior metrics, we assume the model always correctly predicts the protected attribute $A$.) However, as we discuss in the next section, there are some properties of bias amplification that this metric is not able to capture: for example, it does not distinguish between errors in predicting the protected attribute versus errors in predicting the task. Thus we introduce a new metric (last graph of Fig. 2) which maintains the desirable properties from Zhao et al. (2017) while including a number of innovations.

## 3.2 Shortcomings of BiasAmp$_{\text{MALS}}$

Despite the advantages we just documented about BiasAmp$_{\text{MALS}}$ (Eqn. 1) and its ability to distinguish under- and overclassifications of training correlations, this metric also suffers from a number of shortcomings. To ground our discussion, we will work directly with the model outputs released by Zhao et al. (2017) from their Conditional Random Field (CRF) model on COCO (Lin et al., 2014), which has predictions for gender and objects detected for each image.

### 3.2.1 Non-binary attributes

The first shortcoming is that the metric assumes that the protected attributes are binary, limiting its use: the indicator variable $y(t, a)$ implicitly chooses only one of the attribute $a \in \mathcal{A}$ to be associated with every task $t$. Consider a task $t_0 \in \mathcal{T}$ such that $a_0 \in \mathcal{A}$ is associated with it, but none of the other $a_i \in \mathcal{A}$ are, where $i \neq 0$. In this scenario, diff$(t_0, a_i)$ is only considered when there is one other $a_i$ such that $a_i = \neg a$, since diff$(t, a) = -$diff$(t, \neg a)$. A simple addition of $-$diff for all $a_i$'s when $y$ is 0 ensures that when there are more than two groups, their bias amplification is also counted.

### 3.2.2 Base rates

The second shortcoming of BiasAmp$_{\text{MALS}}$ is the fact that the metric does not take into account the base rates of each attribute. Concretely, when determining in $y(t, a)$ of Eqn. 1 whether the attribute $a$ is correlated with the task $t$, $P(A = a | T = t)$ is compared to $\frac{1}{|\mathcal{A}|}$. However, this assumes that all $a$'s within $\mathcal{A}$ are evenly distributed, which may not be the case. For example, in COCO there are about 2.5x as many men as women, so it would appear that most objects positively correlate with men simply by nature of there being an overrepresentation of men. Consider the object oven; BiasAmp$_{\text{MALS}}$ calculates $P(A = \text{man} | T = \text{oven}) = 0.56 > \frac{1}{2}$ and thus considers this object to be correlated with men rather than women. However, computing $P(A = \text{man}, T = \text{oven}) = 0.0103 < 0.0129 = P(A = \text{man})P(T = \text{oven})$ reveals that men are in fact *not* correlated with oven, and the seeming overrepresentation comes from the fact that men are overrepresented in the dataset more generally. Not surprisingly, the model trained on this data learns to associate women with ovens and underpredicts men with ovens at test time, i.e., $P(\hat{A} = \text{man} | \hat{T} = \text{oven}) - P(A = \text{man} | T = \text{oven}) = -0.10$. BiasAmp$_{\text{MALS}}$ erroneously counts this as *inverse* bias amplification.

### 3.2.3 ENTANGLING DIRECTIONS

Another shortcoming we observe is the inability to distinguish between different *types* of bias amplification. Zhao et al. (2017) discovers that "Technology oriented categories initially biased toward men such as `keyboard` ... have each increased their bias toward males by over 0.100." Concretely, from Eqn. 1 $P(A = \texttt{man}|T = \texttt{keyboard}) = .70$ and $P(\hat{A} = \texttt{man}|\hat{T} = \texttt{keyboard}) = .83$, demonstrating an amplification of bias. However, the *direction* or cause of bias amplification remains unclear: is the presence of man in the image increasing the probability of predicting a keyboard, or vice versa? Looking more closely at the model's disentangled predictions, we see that:

$$P(\hat{T} = \texttt{keyboard}|A = \texttt{man}) = 0.0020 < 0.0032 = P(T = \texttt{keyboard}|A = \texttt{man}) \quad (2)$$

$$P(\hat{A} = \texttt{man}|T = \texttt{keyboard}) = 0.78 > 0.70 = P(A = \texttt{man}|T = \texttt{keyboard}) \quad (3)$$

indicating that keyboards are under-predicted on images with men yet men are over-predicted on images with keyboards. Thus the root cause of this amplification appears to be in the gender predictor rather than the object detector. Such disentanglement allows us to properly focus algorithmic intervention efforts. This also highlights the need to consider the ground truth labels on the hold-out set when measuring bias amplification in addition to the predictions (since when considering only the predictions, it is impossible to decouple the different sources of bias).

### 3.3 THRESHOLD

We also fully replicate the original experiment from Zhao et al. (2017) using BiasAmp$_{\text{MALS}}$ on their model predictions and measure .040. However, we observe that "man" is being predicted at a higher rate (75.6%) than is actually present (71.2%). With this insight, we tune the decision threshold on the validation set such that the gender predictor is well-calibrated to be predicting the same percentage of images to have men as the dataset actually has. When we calculate BiasAmp$_{\text{MALS}}$ on these newly thresholded predictions for the test set, we see bias amplification drop from $0.040$ to $0.001$ just as a result of this threshold change, outperforming even the solution proposed in Zhao et al. (2017) of corpus-level constraints, which achieved a drop to only $0.021$. Fairness can be quite sensitive to the threshold chosen (Chen & Wu, 2020), so careful selection should be done in picking the threshold, rather than using the default of .5. In Fig. 3 we show how the amount of bias amplification, as measured by BiasAmp$_{\text{MALS}}$ and BiasAmp$_{T \to A}$, changes as we vary the threshold, i.e., proportion of people classified to be a man. We can see that when the threshold is chosen to be the one well-calibrated on the validation set rather than the default threshold, bias amplification is measured to be closer to zero for both metrics. From here on out when a threshold is needed, we will pick it to be well-calibrated on the validation set. Although we do not take this approach, one could also imagine integrating bias amplification across proportion in order to have a threshold-agnostic measure of bias amplification, similar to what is proposed by Chen & Wu (2020). We do not do this in our experiments because at deployment time, it is often the case that discrete predictions are required.

## 4 BIASAMP$_{\to}$

Now we present our metric, BiasAmp$_{\to}$, which retains the desirable properties of BiasAmp$_{\text{MALS}}$, while addressing the shortcomings noted in the previous section. To account for the need to disentangle the two possible directions of bias amplification (Sec. 3.2.3) the metric consists of two values, BiasAmp$_{A \to T}$ corresponding to the amplification of bias resulting from the protected attribute influencing the task prediction, and BiasAmp$_{T \to A}$, corresponding to the amplification of bias resulting from the task influencing the protected attribute prediction. Concretely, the metric is defined as:

$$\text{BiasAmp}_{\to} = \frac{1}{|\mathcal{A}||\mathcal{T}|} \sum_{t \in \mathcal{T}, a \in \mathcal{A}} y(t,a)\Delta(t,a) + (1 - y(t,a))(-\Delta(t,a)) \quad (4)$$

$$\text{where } y(t,a) = \mathbb{1}\left[P(T = t, A = a) > P(T = t)P(A = a)\right] \quad (5)$$

$$\Delta(t,a) = \begin{cases} P(\hat{T} = t|A = a) - P(T = t|A = a) & \text{if measuring } A \to T \\ P(\hat{A} = a|T = t) - P(A = a|T = t) & \text{if measuring } T \to A \end{cases} \quad (6)$$

Eqn. 4 generalizes BiasAmp$_{\text{MALS}}$ to measure the amplification of both positive and negative correlations between task $t$ and attribute $a$, depending on $y(t,a)$, when the attributes are non-binary,

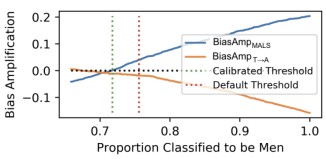

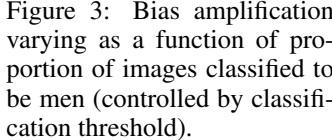

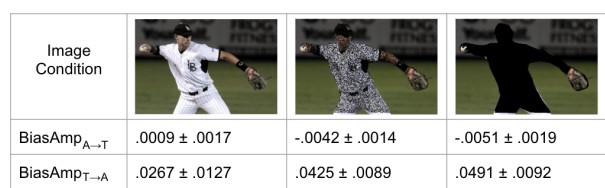

| Image Condition | | | |
|---|---|---|---|
| BiasAmp$_{A \to T}$ | .0009 ± .0017 | -.0042 ± .0014 | -.0051 ± .0019 |
| BiasAmp$_{T \to A}$ | .0267 ± .0127 | .0425 ± .0089 | .0491 ± .0092 |

Figure 3: Bias amplification varying as a function of proportion of images classified to be men (controlled by classification threshold).

Figure 4: BiasAmp$_\to$ changes as a result of three conditions of images: full, noisy person mask, full person mask. As less of the person is visible, A→T decreases from less human attribute visual cues to bias the task prediction. T→A increases because the model must rely on visual cues from the task to predict the attribute.

as discussed in Sec. 3.2.1. Eqn. 5 identifies the direction of correlation of attribute $a$ with task $t$, properly accounting for base rates as described in Sec. 3.2.2. Finally, Eqn. 6 decouples the two possible directions of bias amplification as in Sec. 3.2.3. Since values may be negative, reporting the aggregated bias amplification value could obscure task-attribute pairs that exhibit strong bias amplification; thus, disaggregated results per pair can be returned for a more detailed diagnosis.

Although in much of the examples we have and will look at, $\mathcal{A} = \{\text{woman}, \text{man}\}$, this formulation allows for any attribute set to be defined, including intersectional identities. This is achieved by having $\mathcal{A}$ encompass the cross-product of possible attributes, for example $\mathcal{A} = \{\text{Black woman}, \text{Black man}, \text{white woman}, \text{white man}\}$.

We introduce a scenario for validating the decoupling aspect of our metric, that simultaneously serves as inspiration for an intervention approach to mitigating bias amplification. We use a baseline amplification removal idea of applying segmentation masks (noisy or full) over the people in an image to mitigate bias stemming from human attributes (Wang et al., 2019). We train on the COCO classification task a VGG16 (Simonyan & Zisserman, 2014) model pretrained on ImageNet (Russakovsky et al., 2015) to predict objects and gender, with a Binary Cross Entropy Loss over all outputs, and measure BiasAmp$_{T \to A}$ and BiasAmp$_{A \to T}$; we report 95% confidence intervals for 5 runs of each scenario. In Fig. 4 we see, as expected, that as less of the person is visible, A→T decreases because there are less human attribute visual cues to bias the task prediction. On the other hand, T→A increases because the model must lean into task biases to predict the person's attribute. However, we can also see from the overlapping confidence intervals that this technique of bias amplification mitigation does not appear to be particularly robust; we continue a discussion of this phenomenon in Sec. 5.2. Further mitigation techniques are outside of our scope, but we look to works like Singh et al. (2020); Wang et al. (2019); Agarwal et al. (2020).

## 5 ANALYSIS AND DISCUSSION

We now discuss some of the normative issues surrounding bias amplification, starting in Sec. 5.1 with the existence of T→A bias amplification, which implies the prediction of sensitive attributes; in Sec. 5.2 about the need for confidence intervals to make robust conclusions; and in Sec. 5.3 about scenarios in which the original formulation of bias amplification as a desire to match base correlations may not be what is actually wanted.

### 5.1 T → A BIAS AMPLIFICATION

If we think more deeply about these bias amplifications, we might come to a normative conclusion that $T \to A$, which measures sensitive attribute predictions conditioned on the tasks, should not exist in the first place. There are very few situations in which predicting sensitive attributes makes sense (Scheuerman et al., 2020; Larson, 2017), so we should carefully consider if this is strictly necessary for target applications. For the image domains discussed, by simply removing the notion of predicting gender, we trivially remove all $T \to A$ bias amplification. In a similar vein, there has been great work done on reducing gender bias in image captions (Hendricks et al., 2018; Tang et al., 2020), but it is often focused on targeting $T \to A$ amplification rather than $A \to T$. When

**Baseline**: a *man* standing in front of a market selling bananas **Equalizer**: a woman in a *red dress* is holding an *umbrella* 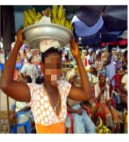

**Baseline**: a *man* riding a motorcycle with a dog on the back **Equalizer**: a woman sitting on a *bench* next to a dog 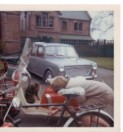

**Baseline**: a *man* brushing his teeth with a tooth brush **Equalizer**: a woman holding a *glass of wine* in her hand 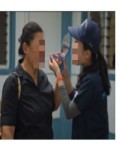

**Baseline**: a *man* and a baby elephant standing in the water **Equalizer**: a woman in a *bikini* standing next to a *dog* 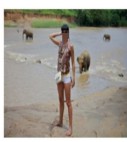

Figure 5: Illustrative captions from the Equalizer model (Hendricks et al., 2018), which decreases $T \rightarrow A$ bias amplification in these captions from the Baseline, but inadvertently increases $A \rightarrow T$. Green underlined words are correct, and red italicized words are incorrect. In the images shown, the Equalizer improves on the Baseline for the gendered word, but also introduces additional biased errors in the captions.

disentangling the directions of bias, we find that the Equalizer model (Hendricks et al., 2018), which was trained with the intention of increasing the quality of gender-specific words in captions, inadvertently increases $A \rightarrow T$ bias amplification for certain tasks. In Fig. 5 we see examples where the content of the Equalizer's caption exhibits bias coming from the person's attribute. Even though the Equalizer model reduces $T \rightarrow A$ bias amplification in these images, it inadvertently increases $A \rightarrow T$. It is important to disentangle the two directions of bias and notice that while one direction is becoming more fair, another is actually becoming more biased. Although this may not always be the case, depending on the downstream application, perhaps this is a setting in which we could consider simply replacing all instances of gendered words like "man" and "woman" in the captions with "person" to trivially eliminate $T \rightarrow A$, and focus on $A \rightarrow T$ bias amplification. Specifically when gender is the sensitive attribute, Keyes (2018) thoroughly explains how we should carefully think about why we might implement Automatic Gender Recognition (AGR), and avoid doing so.

On the other hand, sensitive attribute labels, ideally from self-disclosure, can be very useful. For example, these labels are necessary to measure $A \rightarrow T$ amplification, which is important to discover, as we do not want our prediction task to be biased for or against people with certain attributes.

## 5.2 LACK OF CONSISTENCY IN BIAS MEASUREMENT

Evaluation metrics, ours' included, are specific to each model on each dataset. Under common loss functions such as Cross Entropy Loss, some evaluation metrics, like average precision, are not very sensitive to random seed. However, we noticed that bias amplification, along with other fairness metrics like FPR difference, often fluctuates greatly across runs. Because the loss functions that machine learning practitioners tend to default to using are proxies for accuracy, it makes sense that the various local minima, while equal in accuracy, are not necessarily equal in terms of other measurements. The phenomena of differences between equally predictive models has been termed the Rashomon Effect (Breiman, 2001), or predictive multiplicity (Marx et al., 2020).

Thus, like previous work (Fisher et al., 2019), we urge transparency, and advocate for the inclusion of confidence intervals. To illustrate the need for this, we look at the facial image domain of CelebA (Liu et al., 2015), defining the two tasks of classifying "big nose" and "young" as our $T$, and treating the gender labels as our attribute, $A$. Note that we are not going to classify gender, for reasons raised in Sec. 5.1, so we only measure $A \rightarrow T$ amplification. For these tasks, women are more correlated with no big nose and being young, and men with big nose and not being young. We examine two different scenarios, one where our independent variable is model architecture, and another where it is the ratio between number of images of the majority groups (e.g., young women and not young men) and minority groups (e.g., not young women and young men). By looking at the confidence intervals, we can determine which condition allows us to draw reliable conclusions about the impact of the variable on bias amplification.

For model architecture, we train 3 models pretrained on ImageNet (Russakovsky et al., 2015) across 5 runs: ResNet18 (He et al., 2016), AlexNet (Krizhevsky et al., 2012), and VGG16 (Simonyan & Zisserman, 2014). Training details are in Appendix A.3. In Fig. 6 we see from the confidence intervals that while model architecture does not result in differing enough of bias amplification to conclude anything about the relative fairness of these models, across-ratio differences are significant enough to draw conclusions about the impact of this ratio on bias amplification. We encourage researchers to include confidence intervals so that findings are more robust to random fluctuations.

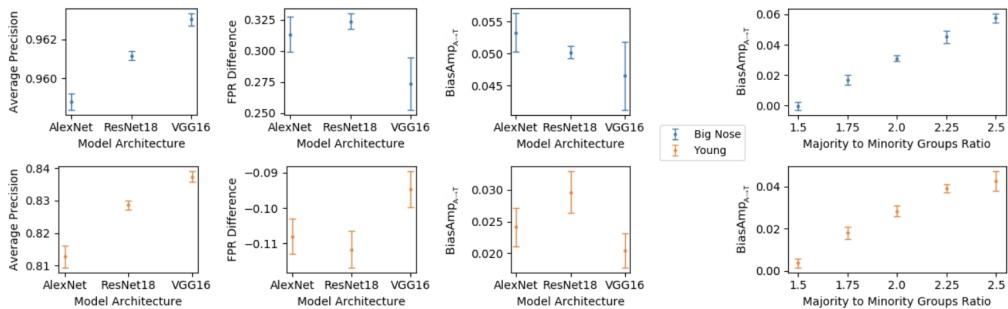

Figure 6: We investigate the consistency of various metrics by looking at 95% confidence intervals as we manipulate two independent variables: model architecture (left three graphs), and majority to minority groups ratio (right graph). The top row is for the attribute of "big nose", and bottom row is for "young." For model architecture, across 5 runs, the accuracy measure of average precision retains a consistent ranking across models, but two different fairness measures (FPR difference and $A \rightarrow T$ bias amplification) have overlapping intervals. This does not allow us to draw conclusions about the differing fairness of these models. However, across-ratio differences in bias amplification are significant enough to allow us to draw conclusions about the differing levels of fairness.

| Base Correlation Source | BiasAmp$_{T \rightarrow A}$ | BiasAmp$_{A \rightarrow T}$ |
|---|---|---|
| Uniform (2 pronouns) | .1875 | .0156 |
| Uniform (3 pronouns) | .1250 | .0104 |
| Wikipedia (2 pronouns) | .2457 | .0088 |
| 2016 U.S. Labor Force (WinoBias) (2 pronouns) | -.0747 | -.0017 |

Table 1: BiasAmp$_{\rightarrow}$ for different base correlation sources.

## 5.3 Considerations for Uncertain Prediction Problems

A property of bias amplification is that it does not conflict with having perfect accuracy. However, in turn, such a model with perfect accuracy would exactly replicate the correlations present in the training data. In this section we will discuss two cases that challenge this desire for matching training correlations: in the first, there will be no ground-truth labels from which to lift these correlations, and in the second, our goal is actually to not match the training correlations.

**No Ground Truth.** When we don't have labels to derive training correlation rates from, bias amplification becomes harder to measure. Consider the fill-in-the-blank NLP task, where there is no ground-truth for how to fill in a sentence. Given "The [blank] went on a walk", a variety of words could be equally suitable. Therefore, to measure bias amplification in this setting, we can manually set the base correlations, e.g., $P(T = t | A = a), P(A = a | T = t)$. To see the effect that adjusting base correlations has, we test the bias amplification between occupations and gender pronouns, conditioning on the pronoun and filling in the occupation and vice versa. In Table 1, we report our measured bias amplification results on the FitBERT (Fill in the blanks BERT) (Havens & Stal, 2019; Devlin et al., 2019) model using various sources as our base correlation of bias from which amplification is measured. The same outputs from the model are used for each set of pronouns, and the independent variable we manipulate is the source of: 1) equality amongst the pronouns (using 2 and 3 pronouns), 2) co-occurrence counts from English Wikipedia (one of the datasets BERT was trained on), and 3) WinoBias (Zhao et al., 2018) with additional information supplemented from the 2016 U.S. Labor Force Statistics data. Details are in Appendix A.4. It is interesting to note that relative to U.S. Labor Force data on these particular occupations, FitBERT actually exhibits no bias amplification. For the occupation of hairdresser, the Labor statistics are biased at 92% women while FitBERT is at 80%, reflecting in fact a reduction in bias amplification. This demonstrates the importance of setting appropriate base correlations, because picking one that already exhibits strong amounts of bias will only flag models that further amplify this. In the next section we discuss another manifestation of this phenomenon, where the training correlation itself would be misleading to compare to, because of the strong amount of bias it contains.

**Future Outcome Prediction.** Next, we examine the risk prediction setting, where matching the base correlations may not be our desired goal. The labels here do not represent an objective ground truth because they: 1) suffer from problems like historical and selection bias (Suresh & Guttag, 2019; Olteanu et al., 2019; Green, 2020), and 2) will be used for prediction of a future event for which no one knows the answer. We will put aside the significant problems stemming from the first point for this current discussion, and focus on the latter. In this section, when using the word "predict" we will revert from the machine learning meaning and adopt the colloquial sense of the word, as in the forecast of a future event. What we had called object prediction, we will now call object labeling.

Consider bias amplification in the context of a domain like risk prediction relative to previous domains looked at, such as object detection. The difference is not in tabular versus vision, but rather in prediction versus labeling. The notion of "ground-truth" doesn't quite exist the way we might think about it, because given the input features that define a particular person, one could imagine an individual with these features who does recidivate, and one who does not (Barocas et al., 2019). The training label is just how someone with these input features once acted, but is not necessarily a rigid indicator of how someone else with these features will act. On the other hand, in object labeling the labels are very much the ground-truth, and thus bias amplification is a reasonable metric to gauge fairness by. However, we do not know this for risk prediction, and thus, matching the training correlations should not be our intended goal (Wick et al., 2019). In Fig. 7 we show the metrics

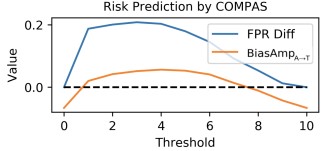

Figure 7: How the fairness metrics of BiasAmp$_{A \to T}$ and FPR difference vary as we change the threshold for COMPAS's risk prediction.

of BiasAmp$_{A \to T}$ and FPR difference measured on COMPAS predictions (Angwin et al., 2016) at different thresholds, only looking at two racial groups we will call 1 and 2. It might be surprising to see that at certain thresholds, bias amplification is 0 or negative. What this means is that Group 1, which is more correlated with the target label than Group 2, has a (predicted rate - base rate) value that is smaller than Group 2's. If we were looking at bias amplification as a way to measure fairness in this domain, this is not a situation where our metric is misreporting, but rather a misspecification of what we are looking for. Like every other fairness metric, ours only captures one perspective, which is that of not amplifying already present biases. It does not require a correction for these biases. As we can see from the continued existence of a difference in FPR's between the two groups, there is no threshold that could be picked at which this disparity does not exist (except at the trivial cases of a 0 or 10 threshold), even if there does exist thresholds where bias is not being amplified.

For each application, different metrics of fairness are more or less applicable, and BiasAmp$_{\to}$ is no different. It is crucial that we think thoughtfully when deciding how to evaluate a model. In previous applications we imagined the direction of systemic bias to be captured by the training data and thus we lifted base correlations of bias from there. However, one could imagine a similarly manual intervention as was done for NLP on other tasks like risk prediction, where a domain expert verifies the direction of bias determined by the training set, and even sets the base correlations.

## 6 CONCLUSION

In this paper, we take a deep dive into the measure of bias amplification. We introduce a new metric, BiasAmp$_{\to}$, and through the use of this metric and its directional components, diagnosing models will provide more actionable insights. Additionally, we discuss normative considerations, such as thinking carefully about why we might be performing sensitive attribute prediction, incorporating confidence intervals as the norm when reporting fairness metrics, and exercising care when determining which fairness metrics are applicable, and what assumptions they are encoding.

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

# A  APPENDIX

## A.1  ADDITIONAL METRIC DETAILS

We provide additional details here about $\text{BiasAmp}_{\rightarrow}$, as defined in Sec. 4.

In practice the indicator variable, $y(t, a)$, is computed over the statistics of the training set, whereas everything else is computed over the test set. The reason behind this, is that the direction of bias is determined by the existing biases in the training set. Additionally, for computing the integration across all thresholds, we sparsely sample from all of the probabilities outputted to compute an approximation.

Comparisons of the values outputted by $\text{BiasAmp}_{\rightarrow}$ should only be done relatively. In particular, within one of the directions at a time, either $A \rightarrow T$ or $T \rightarrow A$, on one dataset. In particular, comparing $A \rightarrow T$ to $T \rightarrow A$ directly is not a signal as to which direction of amplification is stronger.

## A.2  WALKING THROUGH THE EQUATIONS FOR FIGURE 1 SCENARIO

In this section we concretely write out the equations for $\text{BiasAmp}_{\text{MALS}}$ and $\text{BiasAmp}_{\rightarrow}$ for the scenario shown in Fig. 1, to better clarify what each metric captures. As a reminder, in this scenario $\mathcal{A} = \{a_0, a_1\}$ and $\mathcal{T} = \{t_0, t_1\}$, and $a_0$ is correlated with $t_0$, and $a_1$ with $t_1$.

$$\text{BiasAmp}_{\text{MALS}} = \tag{7}$$

$$\frac{1}{2} \sum_{t=1}^{2} \sum_{a=1}^{2} \mathbb{1}\left( P(A_a = 1 | T_t = 1) > \frac{1}{2} \right) \left( P(\hat{A}_a = 1 | \hat{T}_t = 1) - P(A_a = 1 | T_t = 1) \right) \tag{8}$$

$$= \frac{1}{2}\Bigg[ 1 \times \left( P(\hat{A}_0 = 1 | \hat{T}_0 = 1) - P(A_0 = 1 | T_0 = 1) \right) + \tag{9}$$

$$0 \times \left( P(\hat{A}_1 = 1 | \hat{T}_0 = 1) - P(A_1 = 1 | T_0 = 1) \right) + \tag{10}$$

$$0 \times \left( P(\hat{A}_0 = 1 | \hat{T}_1 = 1) - P(A_0 = 1 | T_1 = 1) \right) + \tag{11}$$

$$1 \times \left( P(\hat{A}_1 = 1 | \hat{T}_1 = 1) - P(A_1 = 1 | T_1 = 1) \right) \Bigg] \tag{12}$$

$$= \frac{1}{2}\Bigg[ \left( P(\hat{A}_0 = 1 | \hat{T}_0 = 1) - P(A_0 = 1 | T_0 = 1) \right) \tag{13}$$

$$+ \left( P(\hat{A}_1 = 1 | \hat{T}_1 = 1) - P(A_1 = 1 | T_1 = 1) \right) \Bigg] \tag{14}$$

For $\text{BiasAmp}_{\rightarrow}$, our equation simplifies in the case of discrete predictions as follows:

$$\text{BiasAmp}_{A \rightarrow T} = \frac{1}{4}\Bigg[ \left( P(\hat{T}_0 = 1 | A_0 = 1) - P(T_0 = 1 | A_0 = 1) \right) \tag{15}$$

$$- \left( P(\hat{T}_0 = 1 | A_1 = 1) - P(T_0 = 1 | A_1 = 1) \right) \tag{16}$$

$$- \left( P(\hat{T}_1 = 1 | A_0 = 1) - P(T_1 = 1 | A_0 = 1) \right) \tag{17}$$

$$+ \left( P(\hat{T}_1 = 1 | A_1 = 1) - P(T_1 = 1 | A_1 = 1) \right) \Bigg] \tag{18}$$

$$\tag{19}$$

$$\text{BiasAmp}_{T \to A} = \frac{1}{4} \Bigg[ \Big( P(\hat{A}_0 = 1 | T_0 = 1) - P(A_0 = 1 | T_0 = 1) \Big) \tag{20}$$

$$- \Big( P(\hat{A}_0 = 1 | T_1 = 1) - P(A_0 = 1 | T_1 = 1) \Big) \tag{21}$$

$$- \Big( P(\hat{A}_1 = 1 | T_0 = 1) - P(A_1 = 1 | T_0 = 1) \Big) \tag{22}$$

$$+ \Big( P(\hat{A}_1 = 1 | T_1 = 1) - P(A_1 = 1 | T_1 = 1) \Big) \Bigg] \tag{23}$$

$$\tag{24}$$

### A.3 Details and Experiment from Lack of Consistency in Bias

For the models we trained in Sec. 5.2, we performed hyperparameter tuning on the validation set, and ended up using the following: ResNet18 had a learning rate of .0001, AlexNet of .0003, and VGG16 of .00014. All models were trained with stochastic gradient descent, a batch size of 64, and 10 epochs.

### A.4 Details on Measuring Bias Amplification in FitBERT

Here we provide additional details behind the numbers presented in Tbl. 1 in Sec. 5.3.

As noted, and done, by Liang et al. (2020), a large and diverse corpus of sentences is needed to sample from the large variety of contexts. However, that is out of scope for this work, where we simply use 2 sentences: "[he/she/(they)] is a(n) [occupation]" or "[he/she/(they)] was a(n) [occupation]" to test.

When calculating the amount of bias amplification when the base rates are equal, we picked the direction of bias based on that provided by the WinoBias dataset. In practice, this can be thought of as setting the base correlation, $P(A = a | T = t)$ for a men-biased job like "cook" to be $.5 + \epsilon$ for "he" and $.5 - \epsilon$ for "she" when there are two pronouns, and $.33 + \epsilon$ for "he" and $.33 - \epsilon$ for "she" and "they", where in practice we used $\epsilon = 1\text{e}{-}7$. This ensures that the indicator variable, $y(t, a)$ from Eq. 5, is set in the direction fo the gender bias, but the magnitudes of $\Delta(t, a)$ from Eq. 6 are not affected to a significant degree.

To generate a rough approximation of what training correlation rates could look like in this domain, we look to one of the datasets that BERT was trained on, the Wikipedia dataset. We do so by simply counting the cooccurrences of all the occupations along with gendered words such as "man", "he", "him", etc. There are flaws with this approach because in a sentence like "She went to see the doctor.", the pronoun is in fact not referring to the gender of the person with the occupation. However, we leave a more accurate measurement of this to future work, as our aim for showing these results was more for demonstrative purposes illustrating the manipulation of the correlation rate, rather than in rigorously measuring the training correlation rate.

We use 32 rather than 40 occupations in WinoBias Zhao et al. (2018), because when we went to the 2016 U.S. Labor Force Statistics data (of Labor Statistics, 2016) to collect the actual numbers of each gender and occupation in order to be able to calculate $P(T = t | A = a)$, since WinoBias only had $P(A = a | T = t)$, we found 8 occupations to be too ambiguous to be able to determine the actual numbers. For example, for "attendant", there were many different attendant jobs listed, such as "flight attendants" and "parking lot attendant", so we opted rather to drop these jobs from the list of 40. The 8 from the original WinoBias dataset that we ignored are: supervisor, manager, mechanician, CEO, teacher, assistant, clerk, and attendant. The first four are biased towards men, and the latter four towards women, so that we did not skew the distribution of jobs biased towards each gender.

