# OpenReview forum: "A Technical and Normative Investigation of Social Bias Amplification"
_ICLR.cc/2021/Conference — Reject_

### Official Review · AnonReviewer1 · 2020-10-15
**An insightful perspective about bias amplification backed with sound metrics and analysis**

**Rating:** 7
**Confidence:** 4

**Review:**

Summary:
The paper proposes a new metric for quantification of bias amplification that aids in interpretability of models. In particular, the paper offers useful insights concerning the meaning of bias amplification across varying contexts or use cases and prescribes the use of confidence intervals for better validation of various fairness metrics.

Positives:

1. Technical Sophistication
The paper throughly examines the shortcomings of a previous metric used for computing bias amplification and proposes a new metric that overcomes the limitation of prior metric while still retaining the desirable properties. In particular, the authors clearly state the benefits of their method as described in Sec 3.4.
2. Presentation:
There is sufficient clarity in the paper facilitating easy comprehension.
3. Experiments and analysis:
This is the biggest strength of the paper. A thorough investigation makes this paper compelling. The authors also discuss the limitations of the proposed metric. The running example (fig 1 ) considered in the paper is interesting, especially given that such stereotypes are not considered often.

Concerns:

1. The recommendation for confidence interval is justifiable, but lacks a proper grounding.
2.  The authors use causal notations for bias amplifications with the arrows from A to T or vice versa. While it is ok in terms of notation, it raises the question of how causal claims can be justified by merely thresholding  as opposed to interventions . In this context, more elaboration of the metric (Sec 3.3) would be useful.

Overall comments:

I enjoyed reading this paper, and it offered some interesting insights. I believe it will be valuable in analysis of bias across different use cases in a manner that is interpretable.

---

> ### Author Response · Authors · 2020-11-14
> **Response to R1 regarding causal notation**
>
> Thank you very much for your feedback and comments! We are very happy to hear that the comprehensibility of our paper was good, and our experiments provided a good coverage of the span of both the positives and limitations that come from measuring bias amplification. Please see above (in the response to all) regarding your concern about the confidence interval section, which was shared by the other reviewers.
>
> Regarding the causal notation, we agree that it can be confusing (and in fact did consider this when writing up the paper originally but couldn’t quite think of better notation for what we’re trying to convey). For now we added footnote 1 the first time we use the directional notation (page 1), which states “The arrow in BiasAmp_-> is meant to signify the direction that bias amplification is flowing, and not intended to be a claim about causality.” We would greatly appreciate any other suggestions of what would be helpful for clarity.

---

### Official Review · AnonReviewer4 · 2020-10-24
**Lack of Technical Rigor**

**Rating:** 5
**Confidence:** 4

**Review:**

The paper talks about an interesting aspect in algorithmic fairness meaning bias amplification; however, there are couple of concerns that I will raise below:

1. The paper mainly addresses shortcomings of a previously mentioned simple concept, bias amplification, brought up in Zhao et al 2017. In this regards, I am not sure about extensiveness of novelty and contribution of this paper along with its technical rigor. One suggestion would be to add some theoretical analysis maybe in Validating the metric session.

2. Although Zhao et al's paper is a famous paper in NLP domain, not much attention has not been given in the pure algorithmic fairness and using it as a measure along with other well known measures, such as statistical parity or EO. This makes me even more concerned about an extension to this work.

3. The discussion on error bars and their need sounds like an intuitive concept which authors spent a section on it. Other sections could have been made richer experimentally instead of this section maybe.

4. Analysis of benchmark fairness datasets remains unexplored. Mostly vision datasets are explored, but authors could have done some studies on famous fairness benchmark datasets as well. This can introduce bias amplification concept to the fairness community and make it more comparable to other well known fairness measures and more acceptable to the community.

5. There are some claims that I do not find accurate in the paper as follows:
5.1. "By definition, this problem stems from the algorithms, and can not be attributed to the dataset" -> we can not 100% say this! The bias in the first place is coming from data itself in this case!
5.2. "A trait of bias amplification is that it is not at odds with accuracy, unlike many other metrics ..." -> where is the proof for this? need strong evidence for this claim.

Overall, this paper addresses shortcomings in a previous work based on a simple bias amplification metric. The lack of technical rigor and theoretical guaranties for some claims makes this paper not a strong candidate. Inclusion of some fairness benchmark datasets can make this paper more strong.

---

> ### Author Response · Authors · 2020-11-14
> **Response to R4 regarding additional experiments and clarifying statements**
>
> Thank you for your detailed feedback! In addition to some of the general comments above to all, we also wanted to address a few of your specific notes.
>
> *** Concern #1: The need to better validate the metric, as well as
>
> *** Concern #4: The need to explore fairness benchmark datasets
>
> As per your suggestion, we incorporated a few additional analyses into the updated paper:
>
> a) We added an additional experiment of a partial masking scenario to our COCO masking experiment (Sec. 4 and Fig. 4), where we show that as the amount of visual cues available from the person decreases, there is less human attribute to bias task prediction, so A->T correspondingly decreases. On the other hand, T->A increases as less of the person becomes visible, and the model relies on task biases to predict the human attribute.
>
> b) We included a study of image captioning in COCO (Sec 5.1 and Fig 5), which further drives home the crucial need to disentangle the two directions of bias, as we demonstrate examples where a model intended to mitigate T->A amplification is successful, but at the cost of inadvertently increasing instances of A->T bias amplification.
>
> c) We added a study of risk prediction from COMPAS (Sec 5.3 and Fig. 7). Our analysis on COMPAS is intended to be cautionary and make clear the limitations of bias amplification in certain domains, which we believe is just as important to convey as the benefits of this metric in gauging fairness.
>
> *** Concern #2: The impact of Zhao et al. 2017 outside of NLP and the need to expand upon it
>
> Zhao et al’s paper has been cited 318 times thus far; the following are some examples of works outside NLP domain that use this metric:
> - Towards Fairness in Visual Recognition: Effective Strategies for Bias Mitigation: Zeyu Wang et al. (CVPR 2020)
> - Feature-Wise Bias Amplification: Klas Leino et al. (ICLR 2019)
> - Women also Snowboard: Overcoming Bias in Captioning Models (Table 2): Lisa Anne Hendricks et al. (ECCV 2018)
> - An Intersectional Definition of Fairness: James Foulds et al. (arXiv:1807.08362 2018)
>
> We also hope that our work will help to expand the use of bias amplification further into non-NLP domains, for example, computer vision. We discuss in Sec 5.3 how in CV labeling tasks, bias amplification is often going to be exactly the fairness metric that needs to be met.

---

### Official Review · AnonReviewer3 · 2020-10-29
**Paper deals with Bias Amplification aspect of FairML but lacks a more comprehensive study of the metric and discussion of the normative perspectives.**

**Rating:** 5
**Confidence:** 4

**Review:**

The paper builds on the "bias amplification" aspect of fairness in machine learning literature i.e. the tendency of models to make predictions that are biased in a way that they amplify societal correlations. The paper claims three major contributions: a metric, discussion about the dependence of bias measurements on randomness, and a normative discussion about the use of bias amplification ideas in different domains.

Overall I find the metric as the only major contribution of the paper, and below I will explain why.

The BiasAmp metric makes a significant contribution in terms of fixing the drawbacks of the previously proposed metric from Zhao et al. 2017(BiasAmp_MALS). It would be more effective if the work also included a study such as Zhao et al demonstrating how to mitigate the bias as measured by the BiasAmp measure.

The discussion around--the usage of error bars because of the Rashomon effect seems incomplete and almost trivial. First, in my opinion, it is indeed necessary to have error bars if there are metrics whose measurements vary across different runs--regardless of whether that measure was being optimized on or not. But a more nuanced idea would be either: (a) usually a fairness metric is used as a guidance of whether a specific model is deployable or not, rather than being a property of a training method unless however, an ensemble of a number models trained by the same method is being used in deployment. (b) Fairness metrics may be used as a model selection criterion: since the fairness metric is not being optimized upon but a proxy of accuracy, it just means that, out of the models with equally high accuracy, we need to choose the model with the least bias (as measured by a trustworthy metric).

I find the discussion around the 'consistency' of different metrics incomplete, where average precision (AP) seems to rank models consistently while BiasAmp and others don't. BiasAmp is a pretty sophisticated metric and because of the use of conditional probabilities of all kinds prone to pitfalls such as Simpson's paradox when measuring bias. I would be curious if it were robustly tested e.g. another experiment such as Fig 4 where the authors control the amount of the bias by tuning at the source of the bias.

The discussion around the use of bias amplification in terms of prediction problems where the outcome is chance-based is a little confusing and it does not provide a fresher perspective of discussion already in the fairness literature around understanding the significance of inherent 'uncertainty' or 'randomness' in application domains other than vision (and probably language). For example, the Fair ML book (Barocas, Hardt, and Narayanan 2017) does mention uncertainty in the context of such applications (on page 34, 56).

Overall, I am not very convinced that the paper should be accepted unless fellow reviewers think strongly otherwise. However, I think the paper has the potential to be a more complete and important contribution with a more comprehensive study around the technical contributions and clearer discussion about the normative contributions.

---

> ### Author Response · Authors · 2020-11-14
> **Response to R3 regarding additional experiments and the discussion of prediction applications**
>
> Thank you for your thoughtful feedback! We addressed your two key concerns (regarding the relative strength of the three different claimed contributions, and the importance of the error bars discussion) above in the response to all reviewers. Here we focus on some additional comments.
>
> *** Suggestions for additional experiments: (1) Including “a study such as Zhao et al demonstrating how to mitigate the bias” and  (2) more robust testing, “e.g. another experiment such as Fig 4 where the authors control the amount of the bias by tuning at the source of the bias.”
>
> The focus of our work is on _measuring_ bias amplification, so mitigation strategies are mostly out of our scope, but we agree that providing at least some insight about mitigating bias would definitely be helpful.
>
> First, we expanded on our previous COCO masking experiment (Sec 3.4 before, Fig. 4 and Sec 4 now). By directly controlling the amount of visual cues available from the person, we are able to influence the amount of bias amplification that flows from A(ttribute) to T(ask). As expected, bias amplification decreases as less of the person is visible during task prediction. We thus also frame this experiment better to convey that this is an example of a baseline intervention technique that could be taken to mitigate bias amplification.
>
> Second, we performed additional analysis on the task of COCO image captioning (Sec 5.1 and Fig 5 now; building on the work of Hendricks et al. ECCV 2018). This analysis further drives home the importance and crucial need to disentangle the two directions of bias. In this scenario, we find that a model that was trained to reduce T->A bias amplification inadvertently increases A->T in certain situations.
>
> We have additionally added pointers to other mitigation work (Singh et al. 2020, Wang et al. 2019, Agarwal et al. 2020) at the end of Sec. 4 for the readers’ reference.
>
>
> *** Concerns regarding the discussion of bias amplification in prediction problems (previous section 5.2, now section 5.3)
>
> Thank you for the suggestions. We made a number of changes to this section. First, as mentioned in the general response, we downplayed the relative importance of this discussion from one of the three key contributions of the paper to being just one of the three parts of the bias mitigation metric analysis. Our intention with this section was not necessarily to offer a fresher perspective than what exists, but rather it felt almost irresponsible to present the metric of bias amplification and all of its merits as a way to measure fairness, without also making clear its limitations and downsides. Second, we expanded the discussion to include a case study looking at COMPAS risk prediction data in order to provide more clarity and concreteness. We empirically show that bias amplification is too relaxed of a fairness constraint in domains like risk prediction since it does not correct for dataset biases. Finally, thank you for the suggestion, we added a citation to the Fair ML book (paragraph 2 of Section 5.3’s “Future Outcome Prediction” subsection). We would welcome any additional thoughts or suggestions about this section.

---

### Author Response · Authors · 2020-11-14
**Response to all reviewers regarding (A) the relative strength of the contributions, (B) the discussion of error bars, and (C) the threshold-invariant metric**

We sincerely thank all reviewers for the very helpful feedback and comments. It appears that all reviewers generally found the topic of the paper important and timely, although R3 and R4 raise a few key places where the paper falls marginally below the ICLR bar. We do our best to summarize and respond to the key concerns here.

*** Concern A: We claimed three contributions in the original submission: (1) introduction of the new bias amplification metric, (2) discussion of Rashomon Effect and need to include error bars when reporting fairness measures, and (3) discussion of the use of bias amplification in uncertain prediction problems. The reviewers, especially R3 and R4, raise concerns about the relative strength of the contributions.

We completely agree that the three contributions carry different weights. We struggled with how to present them in the original submission, and ultimately defaulted to slightly over three pages for contribution (1) in Section 3, and about one page each for contributions (2) and (3) in Sections 4 and 5 respectively. However, following the reviewers’ concerns we decided to restructure the paper. We now explicitly focus on the new metric, and incorporate the other contributions into one unified analysis section. Concretely:
- We rephrased the claimed contributions in the last sentence of the introduction to be “Our key contributions are: 1) proposing a new way to measure bias amplification, addressing multiple shortcomings of prior work and allowing us to better diagnose where a model goes wrong, and 2) providing a technical analysis and normative discussion around the use of this measure in diverse settings, encouraging thoughtfulness with each application.”
- We reorganized the rest of the paper around these claims. Section 3 is now “Existing Fairness Metrics” (previous 3.1 and 3.2). Section 4 introduces our metric along with some experimental validation and directions for mitigation (previous 3.3, 3.4). Section 5 is “Analysis and Discussion” (previous 3.5, 4 and 5), and includes three sections on sensitive attribute prediction, inclusion of confidence intervals, and bias amplification in uncertain prediction problems.

We would greatly appreciate any further feedback regarding this organizational structure.

*** Concern B: In our previous contribution (2), we advocate for the need to include error bars when reporting fairness measures. All reviewers raise concerns about this section, namely that the recommendation is “justifiable, but lacks a proper grounding” (R1), the discussion is “incomplete” (R3), and “sounds like an intuitive concept which authors spent a section on it” (R4)

This makes sense. First, as mentioned above, we have now downplayed this section in the presentation by including it as part of the analysis section (now Sec 5.2) rather than as a high-level contribution on par with the introduction of the new metric, and also shortened it to just half a page of text. Second, we have reworked the framing and figure (Fig. 6 in the new draft) to better convey our main message and takeaway, which is that by including confidence intervals, we can make robust conclusions about what affects the bias amplification of models. Finally, we want to emphasize that despite the inclusion of confidence intervals seeming obvious, we have actually found in the literature that the measurement of a fairness metric across one run is often used as a property of a training method itself, and used to reflect how fair one method is over others. We are happy to provide pointers to back up this claim.

*** Comment C: R3 notes that “usually a fairness metric is used as a guidance of whether a specific model is deployable or not”.

This is not a concern per se but we wanted to respond nonetheless. This is similar to a line of thought we had after submitting. In our submission, we proposed a measure of bias amplification that is invariant to the classification threshold (previous Sec 3.2.3). In the updated draft, we continue to offer a version of our metric that is threshold-agnostic and works by integrating over the proportion of images classified (current Sec 3.3), and this could be used depending on the downstream use-case. However, for many applications, a classification threshold does need to be picked, and these discrete outputs are what are used in the real-world. We want our metric to be practically applicable to those cases and thus we added a slightly modified version of our metric (and updated the numbers accordingly) to work on discrete outcomes rather than integrating across all thresholds. We have included a discussion of how important picking the threshold is in Sec 3.3 and Fig. 3 (which we do through calibration on the validation set). Our original findings do not change as a result of this adjustment.

---

### Decision · Program_Chairs · 2021-01-07
**Final Decision**

**Decision:**

Reject

**Comment:**

The authors study bias amplification [Zhao et al, 2017] and propose an improved metric for measuring it. The authors also discussed normative issues in bias amplification (predicting a sensitive feature), and how to measure amplification when we do not have labels, or where labels correspond to uncertain future events. While the reviewers acknowledged the importance to study bias amplification, normative measures and social context, they raised several important concerns:

(1) limited technical contributions (R3 and R4) -- see reviewers’ concerns that the metric is the only technical contribution; one possible suggestion is to propose a mitigation strategy based on the proposed metric similarly to [Zhao et al, 2017];

(2) ‘the usage of error bars because of the Rashomon effect seems incomplete and almost trivial’ (R3, R4), ‘lacks a proper grounding’ (R1) -- see two suggestions by R3 how to revise; these suggestions have not been discussed in the rebuttal;

(3) empirical evidence lacks a controlled scenario of tuning the bias source to evaluate consistency (suggested by R3) and is limited in the context of algorithmic fairness benchmarks (R4) -- this has been partly addressed in the rebuttal;

(4) normative contributions and broader discussions are oversimplified – see R3’s comments and suggestions on how to better position the paper.

Among these, (3) and (4) did not have a major impact on the decision but would be helpful to address in a subsequent revision. However, (1) and (2) make it very difficult to assess the benefits of the proposed work and were viewed by AC as critical issues.
A general consensus among reviewers and AC suggests, in its current state the manuscript is not ready for a publication. It needs technical strength, more empirical studies and polish to achieve the desired goal. We hope the detailed reviews are useful for revising the paper.